# Antibiofilm Effects of N-Acetyl Cysteine on Staphylococcal Biofilm in Patients with Chronic Rhinosinusitis

**DOI:** 10.3390/microorganisms13092050

**Published:** 2025-09-03

**Authors:** Ana Jotic, Ivana Cirkovic, Dragana Bozic, Katarina Savic Vujovic, Jovica Milovanovic, Miljan Folic, Aleksandar Trivic, Ljiljana Cvorovic, Nemanja Radivojevic

**Affiliations:** 1Clinic for Otorhinolaryngology and Maxillofacial Surgery, University Clinical Center of Serbia, Pasterova 2, 11000 Belgrade, Serbia; jmtmilov@gmail.com (J.M.); mfolic@yahoo.com (M.F.); drcole71.at@gmail.com (A.T.); ljiljana.cvorovic.kcs@gmail.com (L.C.); nemanjardv@gmail.com (N.R.); 2Faculty of Medicine, University of Belgrade, Dr Subotica 4, 11000 Belgrade, Serbia; 3Institute of Microbiology and Immunology, Faculty of Medicine, University of Belgrade, Dr Subotica 1, 11000 Belgrade, Serbia; cirkoviciv@yahoo.com; 4Department of Microbiology and Immunology, Faculty of Pharmacy, University of Belgrade, Vojvode Stepe 450, 11221 Belgrade, Serbia; dragana.bozic@pharmacy.bg.ac.rs; 5Department of Pharmacology, Clinical Pharmacology and Toxicology, Faculty of Medicine, University of Belgrade, Dr Subotica 1, 11000 Belgrade, Serbia; katarinasavicvujovic@gmail.com

**Keywords:** chronic rhinosinusitis, staphylococcal bacterial biofilm, N-acetyl-cysteine

## Abstract

Staphylococcal bacterial biofilm plays an important role in the pathogenesis and bacterial persistence of chronic rhinosinusitis. N-acetyl cysteine (NAC) has an inhibitory role in biofilm formation, suppressing adhesion and matrix production or favoring dispersal of preformed biofilm. The aim of this study was to examine the in vitro effect of NAC on Staphylococcal biofilm formation by bacterial strains isolated from tissue samples of patients with chronic rhinosinusitis with or without nasal polyps (CRSwNP and CRSsNP). Prospective study included 75 patients with CRS. The biofilm-forming capacity of isolated strains was detected by microtiter-plate method and the effects of sub-inhibitory (1/2x, 1/4x, and 1/8x minimal inhibitory concentration, MIC) and supra-inhibitory minimal concentrations (2x, 4x, and 8xMIC) of NAC on biofilm production were investigated. Staphylococcal bacterial strains were isolated in 54 (72%) patients, and the most frequently isolated species were *Staphylococcus aureus* (40.7%). Coagulase-negative *Staphylococci* species were weak producers of biofilm, while *S. aureus* was a strong biofilm producer. Concentration of 3.1 mg/mL (1/2 MIC) was sufficient to completely prevent biofilm formation in 77.8% of the isolates, where 49.6 mg/mL (8xMIC) led to the complete eradication of formed biofilm in 81.5% of the isolates. The subinhibitory and eradication effects were dose- and strain-dependent. There were no significant differences in MIC values between isolates from patients with CRSwNP and CRSsNP isolates. NAC proved to be effective in inhibiting biofilm formation and reducing formed biofilm by Staphylococcal isolates from patients with CRS. A comparable antibiofilm effect was exhibited in both phenotypes of CRS, indicating that NAC’s antibiofilm activity was independent of the underlying clinical phenotype, and more targeted on biofilm matrix components.

## 1. Introduction

Chronic rhinosinusitis (CRS) is a frequent chronic inflammatory disease defined as a prolonged inflammation of the sinonasal mucosa that lasts longer than 12 weeks. Pathogenesis is multifaceted, involving immune dysregulation, microbial biofilms, and structural changes in the sinonasal mucosa, which exacerbates local inflammatory responses. One of the important factors in pathogenesis of CRS is biofilm, a complex microorganism community adhered to the surface and surrounded by an extracellular polymeric matrix composed of polysaccharides, proteins, extracellular DNA, and host-derived components [1,2]. *Staphylococcus aureus* remains the most frequently isolated pathogen from the sinonasal mucosa in CRS, detected in up to 60% by culture-independent methods and up to 80% by genomic studies [3]. *Staphylococcus* spp., including *S. aureus*, can form biofilms that enhance bacterial resistance to environmental changes, antibiotics, clearance mechanisms, and immune responses, contributing to persistent disease and treatment failure [4,5,6,7].

This biofilm matrix represents a protective microenvironment that prevents antibiotic activity by physically obstructing antibiotic diffusion, localizing antibiotic-degrading enzymes, and trapping and inhibiting positively charged antibiotics through electrostatic attraction [8]. Peripheral biofilm cells remain metabolically active and susceptible, while deeper layers have limited oxygen and nutrient access, resulting in slow growth, metabolic inactivity, and the emergence of antibiotic-tolerant persister cells. These dormant subpopulations exhibit elevated minimum biofilm eradication concentrations (MBECs), often several hundred- to thousand-times higher than planktonic minimum inhibitory concentrations (MICs) [8,9]. Additionally, *S. aureus* biofilms frequently give rise to small colony variants (SCVs), phenotypic subpopulations characterized by altered metabolism, intracellular persistence, and reduced susceptibility to antimicrobials [10].

Another mechanism of persistence is biofilm-associated immune evasion. *S. aureus* incorporates coagulase-mediated fibrin into the extracellular polymeric matrix, which masks bacterial surfaces from immune recognition. Also, *S. aureus* secreted immune-evasion proteins that interfere with complement activation [11]. After immune cells are recruited, extracellular killing mechanisms such as reactive oxygen species or protease release are mostly directed against the matrix rather than the bacteria themselves, resulting in collateral tissue damage that exacerbates further inflammation without clearance [3].

Stress responses (stringent, SOS, and RpoS pathways) are activated in biofilm-associated bacteria under conditions of nutrient deprivation, hypoxia, and antibiotic exposure, driving mutagenesis through error-prone DNA polymerases [12,13]. Increased cell-to-cell contact, combined with extracellular DNA and the protective biofilm’s matrix, facilitate horizontal gene transfer at higher rates compared to *S. aureus* planktonic cultures [14]. These processes sustain resistant clones and cultivate population heterogeneity, permitting the coexistence of both low-level and high-level resistant variants within the same biofilm [15].

*S. aureus* biofilms directly affect both innate and adaptive immunity, by upregulating pro-inflammatory cytokines and promoting acquired Th-2 inflammatory response within the sinonasal mucosa [16,17,18]. This may additionally explain why Staphylococcal biofilms are strongly associated with disease persistence and treatment failure in both neutrophilic and eosinophilic CRS phenotypes, when compared to biofilms formed by other bacteria [17,19,20].

Biofilm-associated infections are resistant to conventional treatments and require higher concentrations of antimicrobial agents for potential effectiveness, which is not always possible to achieve in clinical settings [21]. This complicates the management of the disease and often leads to recurrent symptoms despite surgical interventions [22,23]. Novel therapeutic strategies are being investigated to optimize treatment outcomes [24].

N-acetylcysteine (NAC) with its mucolytic and antioxidant properties, used in respiratory diseases and infections treatment, proved to be an excellent antibiofilm agent. Previous studies have demonstrated that NAC effectively inhibits biofilm formation in *Staphylococcus* species, particularly *S. aureus* and *Staphylococcus epidermidis* [25,26], supporting its potential use in clinical settings. NAC targets multiple extracellular polymeric matrix components, making its antibiofilm effect independent on bacterial strain [26]. NAC’s effects are dose- and pH-related. Rather than a stand-alone bactericide, NAC acts as a biofilm microenvironment modulator, disrupting and chemically destabilizing the matrix and increasing its permeability. This can enhance antibiotic penetration to the deeper layers of the biofilm, overcoming biofilm-mediated antibiotic resistance which is particularly suited to *S. aureus*’s phenotypic variations [26,27,28]. NAC mitigates oxidative stress by scavenging reactive oxygen species (ROS) and reactive nitrogen species (RNS), while exhibiting anti-inflammatory effects through inhibition of NF-κB activation, reduced cytokine and chemokine production, and suppression of inflammatory gene expression [29]. Additionally, NAC may also modulate the expression of genes involved in biofilm formation and maintenance, influencing the overall architecture and viability of the biofilm [30]. Although NAC has been investigated in the context of respiratory infections, its effect on clinical *Staphylococcus* isolates from CRS remains poorly researched. Since it is widely used and well tolerated, has multiple delivery options, and supports antibiotic-sparing regimens that may decrease antimicrobial resistance, it has excellent potential for clinical implementation in CRS management.

The aim of this study was to examine in vitro effects of NAC on Staphylococcal biofilm formation and previously formed biofilm by bacterial species isolated from tissue samples of patients with chronic rhinosinusitis.

## 2. Materials and Methods

### 2.1. Study Design and Population

A prospective study was conducted from February 2023 until February 2024 at the tertial medical center. The study was approved by the Institutional Ethics Committee (No.29/XII-30, 1 December 2014;17-6/2023, 26 January 2023). All patients signed an informed consent form before being included in the study. The cohort included 75 patients with CRS without nasal polyps (CRSsNP) and CRS with nasal polyps (CRSwNP), as defined by the EPOS 2020 guidelines [31], undergoing endoscopic sinus surgery. Patients underwent a preoperative evaluation with otorhinolaryngologic examination and nasal endoscopy, followed by a computed tomography (CT) of the paranasal sinuses. Lund–Kennedy endoscopic scores were used to assess the pathologic status of sinonasal mucosa in patients intraoperatively. The severity of CRS was evaluated by CT scans using the Lund–Mackay scores. Exclusion criteria were those aged under 18 years old, patients with fungal rhinosinusitis, decreased ciliary function presence, and antibiotic or corticosteroids used in the three weeks preceding surgery.

### 2.2. Tissue Collection

Samples of sinonasal mucosa were obtained intraoperatively and further subjected to detailed microbiological analysis to isolate and identify the microorganisms present.

### 2.3. Isolation and Identification of Bacterial Strains

Isolation and identification of the strains from tissue on selective and differential nutrient media was performed at the Institute of Microbiology and Immunology, Faculty of Medicine, University of Belgrade. Isolates were identified by conventional methods and commercial automated identification systems MALDI-TOF Vitek MS system (BioMérieux, Craponne, France).

### 2.4. Determination of Minimum Inhibitory Concentration

We assessed the antimicrobial activity of N-acetyl-L-cysteine (NAC; AbelaPharm, Belgrade, Serbia) using a standard broth microdilution assay in sterile 96-well plates, following EUCAST recommendations (The European Committee on Antimicrobial Susceptibility Testing. Breakpoint tables for interpretation of MICs and zone diameters. Version 13.0, 2023. http://www.eucast.org (accessed on 31 August 2025)). Bacterial suspensions were prepared in sterile saline, adjusted to a 0.5 McFarland standard (BioMérieux, Craponne, France), and then diluted in Mueller–Hinton broth (MHB; Lab M Limited, Heywood, UK) to a final inoculum of approximately 5 × 10^5^ CFU/mL of microorganisms. NAC was freshly dissolved in MHB and subjected to twofold serial dilutions (1:4–1:64), to working concentrations from 0.62 to 40 mg/mL. Triplicates of each concentration were dispensed into flat-bottom 96-well microtiter plates (ThermoFisher Scientific, Waltham, MA, USA) and inoculated with the standardized bacterial suspension. To monitor growth and metabolic activity, MHB contained 0.05% 2,3,5-triphenyl-2H-tetrazolium chloride (TTC; Sigma-Aldrich, St. Louis, MO, USA) as a redox indicator, which was reduced by cellular dehydrogenases from a colorless state to red, insoluble formazan. Plates were incubated aerobically at 35 ± 1 °C for 18 ± 2 h. The minimum inhibitory concentration (MIC) was defined as the lowest NAC concentration with no visible TTC-related color change after incubation for 18 ± 2 h at 35 ± 1 °C in aerobic conditions. Each run included positive growth controls (inoculated medium without NAC) and negative controls (medium with NAC, no bacteria). All conditions were tested in technical triplicate, and the assay was repeated three times.

### 2.5. Effects of NAC on Biofilm Formation

To test the biofilm-producing capacity, 96-well microtiter plates were used according to Stepanovic et al. [32]. Bacteria were inoculated on tryptic soy agar (BioMérieux, Craponne, France) or chocolate agar (BioMérieux, Craponne, France) and cultivated 18 ± 2 h at 35 ± 1 °C in aerobic/microaerophilic conditions. One colony of each isolate was resuspended in saline to obtain a bacterial suspension corresponding to McFarland standard 0.5 (1.5 × 10^8^ CFU/mL).

Effects of NAC on biofilm formation were tested by making serial dilutions of compounds in the range of 1/2 to 1/8 MIC prepared in tryptic soy broth (TSB; BioMérieux, Craponne, France) supplemented with 1% glucose. For each concentration, 180 μL was dispensed in triplicate into a 96-well microtiter plate, followed by 20 μL of the standardized bacterial suspension added to each well. The positive control of each strain was triplicate of bacteria cultured only in the medium, and two triplicates of medium alone represented the negative control of each plate. To assess NAC-mediated eradication of established biofilms, bacteria were pre-cultivated in TSB for 24 h at 35 ± 1 °C under aerobic or microaerophilic conditions, after which wells were rinsed with sterile phosphate-buffered saline (PBS; pH 7.2). NAC was then prepared in TSB at 2x to 8xMIC in twofold serial dilutions, and 200 µL of each concentration was added in triplicate to wells containing the preformed biofilm. Each plate included positive and negative controls as previously described.

Following a further 24 h incubation at 35 ± 1 °C (aerobic or microaerophilic conditions), plates were washed with PBS, air-dried, fixed with methanol, and stained with 2% crystal violet (Sigma-Aldrich, St. Louis, MO, USA). Bound dye was eluted with 96% ethanol, and biofilm biomass was quantified at optical density of 570 nm (OD_570_) using a Multiskan™ FC microplate reader (ThermoFisher Scientific, Waltham, MA, USA). Biofilm production was classified relative to the cut-off optical density (ODc, defined as the mean OD of the negative control + 3 SD) as category 0 (OD ≤ ODc, no biofilm), category 1 (ODc < OD ≤ 2 × ODc, weak biofilm production), category 2 (2 × ODc < OD ≤ 4 × ODc, moderate biofilm production), and category 3 (OD > 4 × ODc, strong biofilm production).

### 2.6. Statistical Analysis

Descriptive statistics were used to summarize demographics and study variables as counts and percentages. Group differences were examined with the χ^2^ test for categorical data and Student’s *t*-tests for continuous data; when multiple pairwise comparisons were required, post hoc *t*-tests were Bonferroni-adjusted. For nonparametric comparisons, the Kruskal–Wallis test was applied. *p* values < 0.05 were considered statistically significant. All analyses were performed in IBM SPSS Statistics, version 26 (IBM Corp., Armonk, NY, USA).

## 3. Results

### 3.1. Demographic and Clinical Characteristics

The study included 75 patients with chronic rhinosinusitis, 48 (64%) males and 27 (36%) females (*p* < 0.05). The average age of the patients was 45.5 years (from 24 to 75 years). Demographic and clinical characteristics of the patients were shown in Table 1.

Staphylococcal bacterial strains were isolated in 54 (72%) patients. The most frequently isolated pathogen was *S. aureus* in 22 (40.7%), and coagulase-negative *Staphylococci* (CNS) including *S. epidermidis* in 21 (38.9%), *Staphylococcus haemolyticus* in 8 (6.1%) patients. (14.8%), *Staphylococcus warneri* in 2 (3.7%), and *Staphylococcus lugdunensis* in 1 (1.8%) patient. A single bacterial species was isolated in 49, and two different bacterial isolates were detected in 5 patients (Figure 1).

### 3.2. Antimicrobial Activity of NAC

Minimum inhibitory concentrations of NAC were in the range of 2.5 to 10 mg/mL NAC. In *S. aureus* mean MIC was 6.4 mg/mL NAC and in CNS mean MIC was 6.1 mg/mL NAC). The average value of MIC for all isolates was 6.2 mg/mL. The lowest and the highest MIC values for each species were presented in Table 2.

### 3.3. Biofilm Production

The category of biofilm production was calculated in relation to the cut-off value optical density (OD). OD in our samples was 0.065. Calculated values of OD between 0.065 and 0.130 were categorized as weak biofilm production, values between 0.130 and 0.260 were categorized as moderate biofilm production, and OD values above 260 were categorized as strong biofilm production. According to those values, isolate strains were categorized as weak, moderate, and strong biofilm producers. The largest number of isolated strains were weak (21 patients, 38.8%) and moderate (12 patients, 31.5%) producers of biofilm, and 16 isolates (29.7%) were strong biofilm producers. The lowest amount of biofilm was produced by CNS species, *S. haemolyticus* and *S. epidermidis* (category 1), and the highest amount by *S. aureus* (category 2 and 3) (*p* < 0.05). The capacity of strains for in vitro biofilm production isolated from patients with chronic rhinosinusitis depended both on bacterial species and specific strain. Different strains of the same species had different biofilm production capacity, which is presented in Figure 2.

### 3.4. Effect of NAC on Biofilm Formation

Subinhibitory concentrations reduced biofilm formation in all applied concentrations (*p* < 0.05). (Figure 3) The effect was dose-dependent; a concentration of 3.1 mg/mL (1/2 MIC) completely prevented the formation of biofilm in 77.8% of isolates, and a concentration of 1.55 mg/mL (1/4 MIC) in 27.7% of isolates. A concentration of 0.775 mg/mL (1/8 MIC) did not lead to the complete inhibition of biofilm production, although most isolates (57.4%) significantly reduced biofilm production from strong to weak (12 isolates, 22.2%), strong to moderate (4 isolates, 7.4%), or moderate to weak (15 isolates, 27.8%). Based on the Kruskal–Wallis tests, there were no statistically significant differences in MIC values between CRSwNP and CRSsNP groups for inhibition of biofilm formation, across all tested bacterial species and NAC concentrations. The subinhibitory effect also depended on specific bacterial strain (Figure 3).

### 3.5. Effect of NAC on Biofilm Eradication

Supra-inhibitory concentrations reduced the amount of formed biofilm in all applied concentrations (*p* < 0.05) (Figure 4). The effect was dose-dependent, a concentration of 12.4 mg/mL (2xMIC) completely eradicated biofilm in 10 isolates (18.5%) and reduced the amount of formed biofilm in 32 isolates (59.2%). Concentration of 24.8 mg/mL (4xMIC) eradicated biofilm in 23 (42.6%) isolates and reduced the amount of formed biofilm in 25 isolates (46.3%). In the eradication of biofilm, the most effective concentration was again the 49.6 mg/mL (8xMIC), which led to the complete eradication of biofilm in 44 (81.5%) isolates and reduction in the category in all others (*p* < 0.05). The Kruskal–Wallis test results show no statistically significant differences in MIC values between CRSwNP and CRSsNP groups for biofilm eradication, across all tested bacterial species and NAC concentrations. The eradication effect was also strain-dependent (Figure 4).

## 4. Discussion

### 4.1. Staphylococcal Bacterial Biofilms in Chronic Rhinosinusitis

CRS poses a considerable burden on healthcare systems and calls for personalized treatment approaches to address its complex, multifactorial etiology and persistent course [33,34]. Presence of bacterial biofilm in the sinonasal mucosa allows the pathogenic bacteria to persist and survive in the upper respiratory tract [35,36]. *S. aureus* stands out as the most frequently isolated Staphylococcal bacteria in our CRS samples. Its high virulence factors include adhesins that facilitate epithelial attachment, secreted enzymes and toxins, and immune-evasion strategies [37]. On the contrary, *S. epidermidis* was described as dominant commensal in the human nasal microbiome, whose pathogenic contribution in CRS is variable and not always associated with severity of the disease. Biofilm-forming strains of *S. epidermidis* are highly resistant to epithelial antimicrobial peptides (AMPs) and simultaneously stimulate the production of LL-37 and β-defensins, creating a competitive mucosal environment unfavorable for pathogenic *S. aureus* strains [38]. The prevalence of weak and moderate biofilm-forming *S. epidermidis* isolates in our cohort may suggest that their protective potential is diminished compared to commensal strains described in healthy mucosa. As a competitive strain, *S. aureus* proved to be dominant in biofilm formation in our samples. *S. aureus* biofilms are not only structurally more resilient but also stimulate epithelial Nod2/NF-κB and inflammasome pathways, leading to secretion of IL-6, IL-8, CXCL2, and IL-1β/IL-18. These responses promote persistent, type-skewed inflammation, epithelial barrier disruption, and impaired mucociliary clearance in CRS [20].

Although less studied in CRS, other CNS may still play a role in the sinonasal biofilm environment. *S. haemolyticus* possesses notable antibiotic resistance and moderate biofilm capacity, features that could facilitate infection in selected cases [39]. *S. warneri*, while typically a weak biofilm producer with low inherent virulence, may still sustain chronic low-grade inflammation in sinonasal biofilm communities [40]. *S. lugdunensis* displays a more aggressive profile, with robust biofilm formation and virulence features like those of *S. aureus*, suggesting that its presence in CRS could carry greater clinical significance [41]. By isolating these species, we presume that a spectrum of *Staphylococci* may participate in biofilm ecology, and their strain-specific traits require further examination in the context of CRS pathogenesis.

Our findings confirm that the biofilm-forming capacity of Staphylococcal isolates in CRS was both species- and strain-dependent, indicating significant microbiologic heterogeneity. These differences show that not all *Staphylococci* contribute equally to disease chronicity and that biofilm-associated virulence cannot be generalized across species. Instead, biofilm output and pathogenic impact are highly strain-specific, indicating that accurate interpretation of CRS microbiology requires integration of phenotypic testing with genomic characterization.

### 4.2. The Effect of NAC on Staphylococcal Bacterial Biofilms in Chronic Rhinosinusitis

Most of the studies in the literature examined the effect of NAC on various types of reference and clinical strains, not exclusively from patients with chronic sinusitis. Pérez-Giraldo et al., were first to examine the influence of NAC on biofilm formation by 15 clinical isolates of *S. epidermidis*. Inhibition of biofilm formation was dose-dependent and varied from 0.25 to 8 mg/mL, where dose of 1 mg/mL reduced 75% of biomass in most strains [42]. Landini et al., used both *S. aureus* ATCC and clinical stains (MSSA and MRSA), where MIC was ≥16 mg/mL for all *S. aureus* strains. Biofilm formation inhibition was effective with concentrations of NAC between 2.5 and 10 mg/mL, where up to 80% reduction in biomass happened at 10 mg/mL [43]. A group of Turkish authors achieved significant biofilm eradication with higher concentrations of NAC (from 32 to 64 mg/mL, 2–4xMIC) on clinical and reference *S. aureus* and *S. epidermidis* isolates [44]. The result of our study corresponds with those previously published, with MIC varying from 2.5 to 10 mg/mL.

NAC was used for decades in the treatment of chronic respiratory diseases for its mucolytic and antioxidant properties, rather than direct antimicrobial activity [45,46]. According to our data, efficacy of NAC on biofilm formation and preformed biofilm eradication was dose-dependent and strain-dependent. This pattern indicates that NAC affects primarily bacterial and matrix determinants, making it less susceptible to existing resistance mechanisms [25]. The primary mechanism of action involves the intrinsic acidity of NAC, which facilitates the degradation of extracellular DNA (eDNA) through both pH-dependent and potentially pH-independent pathways. The acidity of NAC proved to be directly concentration-dependent, where higher concentrations of NAC result in a lower pH [26]. In the presence of NAC, significant reduction in polysaccharide in the extracellular polymeric matrix (EPM) and bacterial adhesion to surfaces was noted in Staphylococcal biofilm (particularly in MRSA biofilms). By reducing matrix polysaccharides, cleaving extracellular DNA [47], and disrupting thiolated protein cross-links, NAC weakens the matrix protecting the biofilm community. The consequences include shorter diffusion paths and less sequestration for potentially co-administered antimicrobials, collapse of protected, hypoxic niches where slow-growing cells and persisters are located, and diminished matrix–host crosstalk that maintains inflammation [48]. Ultimately, biofilm biomass and maturation, as well as viable bacterial counts, can be significantly reduced, even in mature biofilms [26,42,49,50].

Considering the heterogeneity observed among *Staphylococcus* strains in our study, the observed dose- and strain-specific differences likely reflect which biofilm elements dominate in each isolate. Distinct biofilm architectures of different Staphylococcal species (PIA-dominant versus protein/eDNA-based) and metabolic states create variable barriers, requiring different concentrations for biofilm disruption [51]. This can explain why some strains require higher NAC concentrations for disruption. The absence of differences in MIC values between CRSwNP and CRSsNP isolates for any bacterial species (*p* > 0.05) in our study exhibits a consistent antibiofilm effect independent of the clinical inflammatory phenotype. This further confirms NAC’s matrix-centered mode of action and highlights its therapeutic potential as a useful adjuvant in CRS treatment.

Advantages of this study involve the use of exclusively clinical isolates, which may most accurately mirror the behavior of bacteria in clinical settings. Reference strains, although with consistent properties and reproducibility in experiments, may have different biofilm-forming capacities and may not evolve under specific laboratory conditions like wild-type strains. These characteristics could potentially limit their relevance in biofilm studies and results should be interpreted with care [52,53]. Also, this is the first study that included data obtained from patients with both phenotypes of chronic rhinosinusitis.

This study has several limitations that should be mentioned. The number of included *Staphylococcus* clinical isolates is relatively small, but it remains comparable to or greater than that reported in previous in vitro investigations on NAC’s antibiofilm effects and provides a meaningful contribution to the existing body of evidence. Also, the study was conducted under in vitro conditions, which may not fully replicate the complexity of the sinonasal environment in vivo. Factors such as mucus viscosity, host immune response, epithelial barriers, and microbial diversity may influence the clinical efficacy of NAC and limit direct extrapolation of these findings.

For successful translation of results to clinical settings, the route of administration and local concentration of NAC are crucial for its antibiofilm effect. While NAC is traditionally used orally at doses ranging from 200 to 600 mg, two to three times daily, its systemic bioavailability at sinonasal mucosal surfaces is likely insufficient for effective biofilm disruption. Alternative delivery strategies, including intravenous infusion, nebulization, and direct topical application, have been explored in other settings [54]. Of particular interest are commercially available intranasal formulations of NAC, which may offer a practical and targeted means of delivering high local concentrations to affected mucosa [55]. Such formulations could prove especially valuable for patients with chronic rhinosinusitis, where persistent biofilms play a pathogenic role. Future in vivo studies and clinical trials in larger cohorts with assessment of clinical symptoms are needed to define optimal dosing regimens and confirm therapeutic efficacy in the context of sinonasal biofilm-associated disease.

## 5. Conclusions

NAC proved to be effective in reducing biofilm formation and the amount of preformed biofilm by Staphylococcal isolates from patients with chronic rhinosinusitis. Efficacy of NAC on biofilm formation and preformed biofilm eradication was dose-dependent. The absence of significant differences in NAC MICs between isolates from CRSwNP and CRSsNP indicates that NAC’s antibiofilm activity was independent of the underlying clinical inflammatory phenotype, and more targeted on biofilm matrix components. These findings support adding NAC to topical CRS treatments to disrupt and destabilize biofilm, enhance antibiotic access, and potentially limit recurrence. By testing clinical *Staphylococcus* isolates and assessing both inhibitory and eradication effects, our study highlights NAC as a practical, matrix-active substance with high clinical applicability in CRS treatment.

## Figures and Tables

**Figure 1 microorganisms-13-02050-f001:**
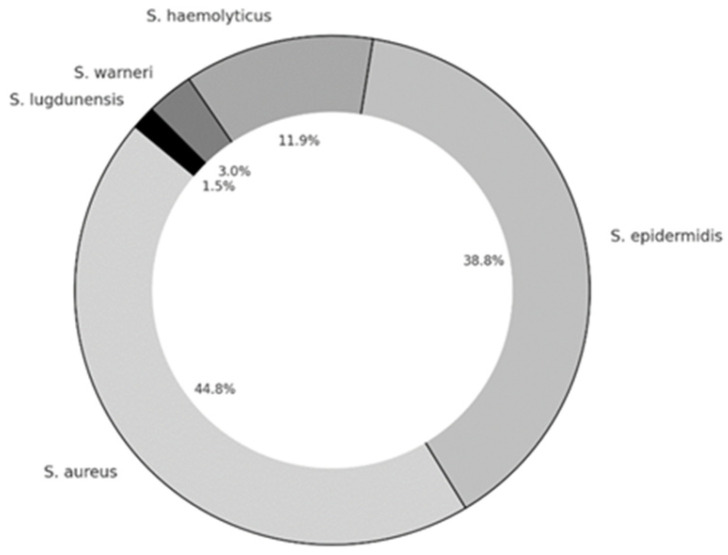
Staphylococcal bacterial isolates in patients with chronic rhinosinusitis.

**Figure 2 microorganisms-13-02050-f002:**
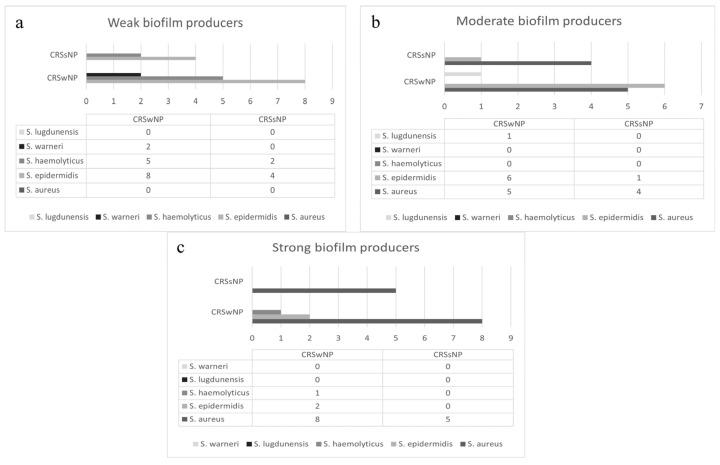
The biofilm production capacity of *Staphylococci* isolated from patients with CRSsNP and CRSwNP, weak biofilm producers (**a**), moderate biofilm producers (**b**) and strong biofilm producers (**c**).

**Figure 3 microorganisms-13-02050-f003:**
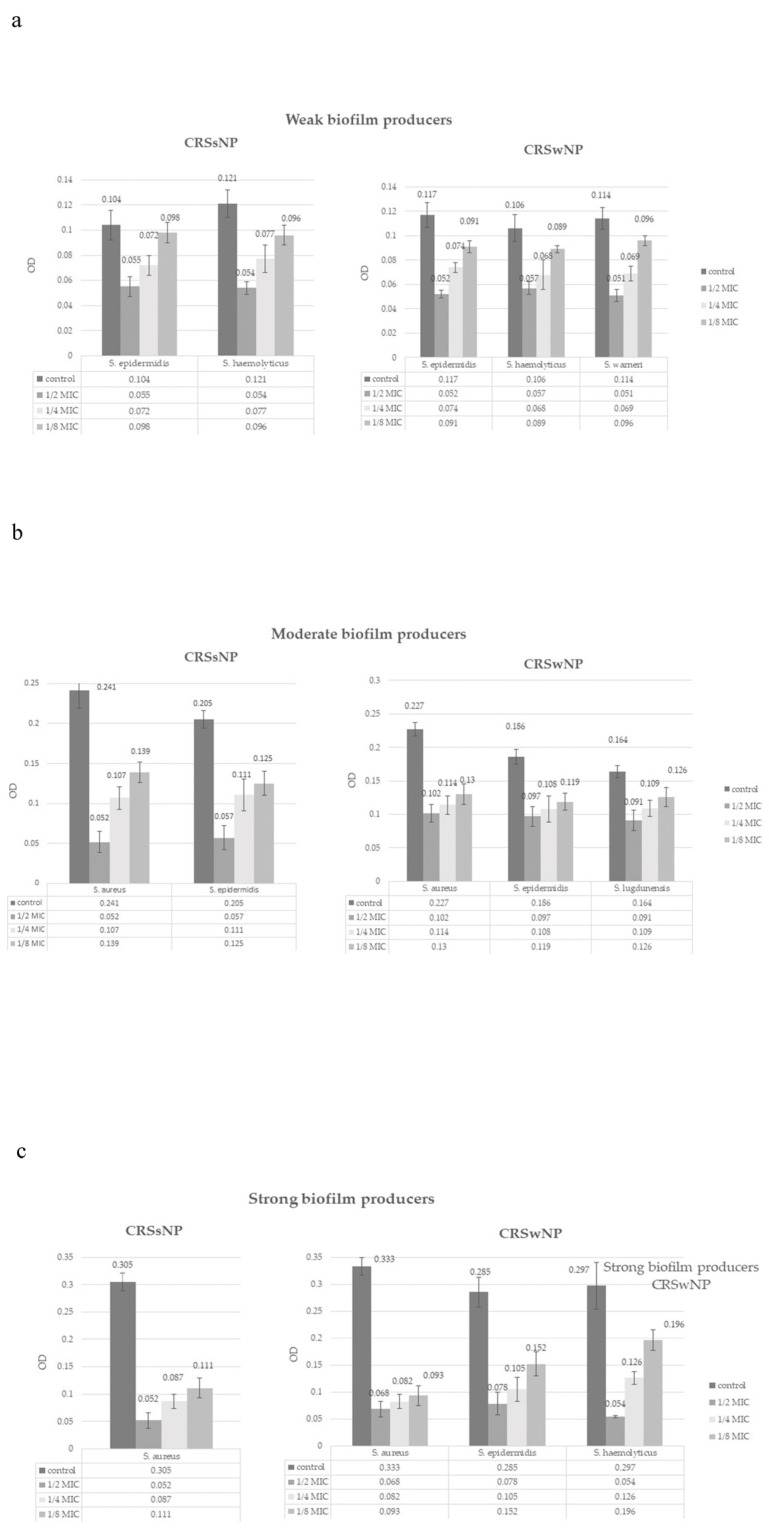
The effect of NAC on biofilm formation in Staphylococcal isolates, (**a**) weak biofilm producers, (**b**) moderate biofilm producers, (**c**) strong biofilm producers.

**Figure 4 microorganisms-13-02050-f004:**
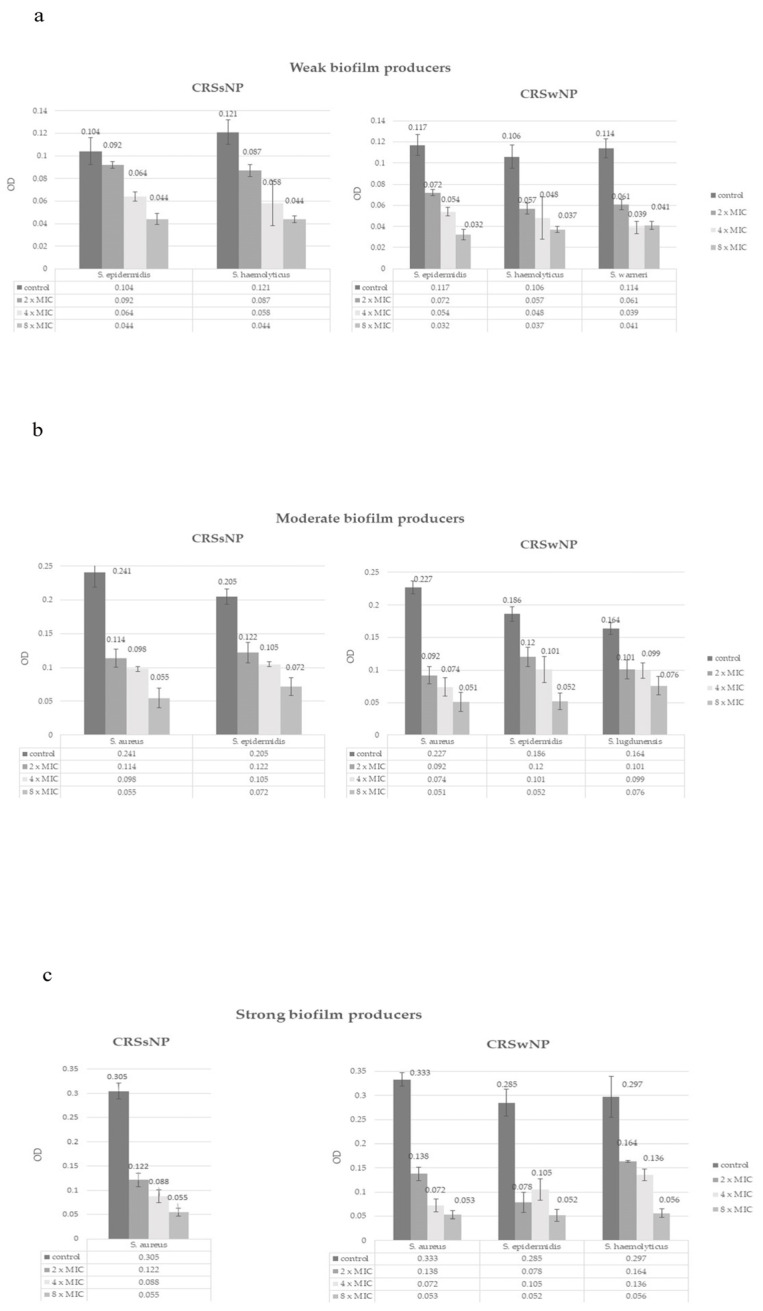
The effect of NAC on eradication of previously formed biofilm in Staphylococcal isolates, (**a**) weak biofilm producers, (**b**) moderate biofilm producers, (**c**) strong biofilm producers.

**Table 1 microorganisms-13-02050-t001:** Demographic and clinical characteristics of the patients with chronic sinusitis.

Sex	
**Male** **Female**	48 (64%) 27 (36%)
**Age (mean ± SD)**	45.5 ± 14.3
**Type of chronic rhinosinusitis *n* (%)**	
**Chronic rhinosinusitis without nasal polyps (CRSsNP)** **Chronic rhinosinusitis with nasal polyps (CRSwNP)**	33 (44%) 42 (64%)
**Lund–MacKay score** **(maximum = 24)**	9 (4–23)

SD—standard deviation.

**Table 2 microorganisms-13-02050-t002:** The lowest and the highest MIC values for Staphylococcal isolates.

Microorganism	Highest MIC (mg/mL)	Lowest MIC (mg/mL)	Mean ± SD MIC (mg/mL)
*Staphylococcus aureus*	10	2.5	6.4 ± 2.8
*Coagulase-negative Staphylococci*	10	2.5	6.1 ± 2.8

## Data Availability

Data set is available for viewing on Open Science Framework (OSF) platform, https://osf.io/ster3/?view_only=440b2121b6d7434abe7f412e1d1038f0 (accessed on 31 August 2025). Detailed data are available on request.

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
