# Peer review of "Antibiofilm Effects of N-Acetyl Cysteine on Staphylococcal Biofilm in Patients with Chronic Rhinosinusitis"

_microorganisms, 2025, doi:10.3390/microorganisms13092050_

Round 1

Reviewer 1 Report

Comments and Suggestions for Authors

This manuscript investigates the inhibitory effect of N-acetylcysteine on Staphylococcal biofilms, a pathogenic bacterium of chronic sinusitis, and completes an in vitro evaluation of NAC's antibacterial capacity. The research has a reasonable purpose and has reference value for practical application, but there are some problems in the writing that need to be improved to make the paper more perfect.

  1. The authors did not adequately present key experimental data in the abstract, only showing the biofilm inhibition rate at a specific MIC ratio, without specifying the numerical values of the MIC and other key data. This makes the conclusion lack sufficient persuasiveness and is not conducive to demonstrating the workload of this experiment.
  2. The authors discuss the purpose of the article in the introduction but neglect to explain the selection of NAC. The rationale for this choice is insufficient. The introduction describes NAC as widely used in the treatment of respiratory infections and has also been studied for its ability to inhibit staphylococci. The authors need to further elaborate on this section to demonstrate the sufficient innovation of the article's purpose and findings and their significant potential for application.
  3. Can the authors provide strain identification information to ensure that the experiments were performed on the target strains? Does the presence of different strains of the target bacteria significantly affect the experimental results?
  4. The authors should add error bars to the graphs to reflect the reliability of the experimental data.
  5. The author's interpretation of the experimental data is relatively simple, only showing the data results without a more in-depth analysis of the mechanism. The author needs to strengthen the analysis and further explore the potential of the experimental data.
  6. The discussion in Section 4.1 is overly cluttered with background information, drawn from experiments conducted by others. This information cannot directly support the mechanism of this study, but serves only as supporting evidence. The authors should reposition this background information in an appropriate place, such as the introduction, or integrate it with the mechanistic analysis of this study to strengthen its persuasiveness.
  7. The discussion section is fragmented and poorly organized. The authors directly present the data and discuss the limitations of the experiment, but lack sufficient discussion of the significance of the results, making it less persuasive. This lack of a good presentation of the experimental results and the weak analysis undermine the value of this paper.
  8. The conclusion is too short; it is concise but not refined. The authors need to further demonstrate the significance of this study here.

Author Response

This manuscript investigates the inhibitory effect of N-acetylcysteine on Staphylococcal biofilms, a pathogenic bacterium of chronic sinusitis, and completes an in vitro evaluation of NAC's antibacterial capacity. The research has a reasonable purpose and has reference value for practical application, but there are some problems in the writing that need to be improved to make the paper more perfect.

  • We thank the reviewer for the insightful and useful comments, allowing us to significantly improve the paper. The answers to the comments are following:

 Comment 1. The authors did not adequately present key experimental data in the abstract, only showing the biofilm inhibition rate at a specific MIC ratio, without specifying the numerical values of the MIC and other key data. This makes the conclusion lack sufficient persuasiveness and is not conducive to demonstrating the workload of this experiment.

Response 1. Numeric values and another important findings were added in the abstract

Comment 2. The authors discuss the purpose of the article in the introduction but neglect to explain the selection of NAC. The rationale for this choice is insufficient. The introduction describes NAC as widely used in the treatment of respiratory infections and has also been studied for its ability to inhibit staphylococci. The authors need to further elaborate on this section to demonstrate the sufficient innovation of the article's purpose and findings and their significant potential for application.

Response 2. The Introduction section was expanded to further elaborate NACs mechanism of action, innovation of action and the translational and application potential. We emphasize NAC’s unique mechanisms of action on biofilm architecture—disruption of extracellular polysaccharides, degradation of eDNA, and reduction of disulfide bonds—that makes it distinct from other substances. Also, according the second reviewer’s comment, some characteristics of the Staphylococcal biofilm were added to better explain how S. aureus biofilms contribute to disease persistence and treatment failure in CRS.

Comment 3. Can the authors provide strain identification information to ensure that the experiments were performed on the target strains? Does the presence of different strains of the target bacteria significantly affect the experimental results?

Response 3. We thank the reviewer for this valuable comment. Species-level identification was performed using MALDI-TOF MS technology, which has been demonstrated to be highly reliable and superior to conventional phenotypic methods for rapid and accurate bacterial identification (Clark et al., Clin Microbiol Rev. 2013;26(3):547–603). All isolates were identified to the species level, and the corresponding results are presented in Section 3.1 and Figure 1.

We agree that different strains of the same species may display variable responses to NAC. This is clearly illustrated in Figures 2, 3, and 4, particularly in the case of S. epidermidis, where strains were classified as weak, moderate, or strong biofilm producers. Our data show that the inhibitory and eradicating effects of NAC are not only dose-dependent, but also influenced by the intrinsic biofilm-forming capacity of individual strains. We are grateful for this observation and will incorporate this clarification into the Results section to strengthen the presentation.

Comment 4. The authors should add error bars to the graphs to reflect the reliability of the experimental data.

Response 4.The graphs were edited, and the error bars were added to the graphs for better presentation of experimental results.

Comment 5. The author's interpretation of the experimental data is relatively simple, only showing the data results without a more in-depth analysis of the mechanism. The author needs to strengthen the analysis and further explore the potential of the experimental data.

Response 5. We appreciate this valuable comment. In the revised version, we have expanded the discussion to provide a deeper mechanistic interpretation of our findings. We now linked the dose- and strain-dependent effects of NAC to differences in biofilm matrix composition (PIA vs protein/eDNA dominance), heterogeneity of bacterial subpopulations, and and degradation of extracellular DNA. We also discuss how these mechanisms may explain variability across isolates and how they translate into potential therapeutic applications in CRS. We think that it highlights the significance of our findings beyond the presentation of raw data.

Comment 6. The discussion in Section 4.1 is overly cluttered with background information, drawn from experiments conducted by others. This information cannot directly support the mechanism of this study, but serves only as supporting evidence. The authors should reposition this background information in an appropriate place, such as the introduction, or integrate it with the mechanistic analysis of this study to strengthen its persuasiveness.

Response 6. Some of the paragraphs of the Discussion section were moved to the Introduction section, and other were edited to better support obtained results.

Comment 7. The discussion section is fragmented and poorly organized. The authors directly present the data and discuss the limitations of the experiment, but lack sufficient discussion of the significance of the results, making it less persuasive. This lack of a good presentation of the experimental results and the weak analysis undermine the value of this paper.

Response 7. We thank the reviewer for this important observation. In the revised manuscript, we have reorganized the Discussion section to improve clarity and flow. Rather than directly repeating the data, we now begin with the broader context of biofilms in CRS, emphasize the novelty of our findings, and then integrate mechanistic explanations of how NAC exerts its antibiofilm effects. We highlight the clinical relevance of our results (particularly the species- and strain-dependent biofilm differences, the independence from CRS phenotype, and the dose-dependent effect of NAC) before addressing limitations and future directions.

Comment 8. The conclusion is too short; it is concise but not refined. The authors need to further demonstrate the significance of this study here.

Response 8. The Conclusion was edited to better demonstrate the significance of the study.

Reviewer 2 Report

Comments and Suggestions for Authors

Recommendation for revision:

Query#1

In the introduction paragraph, the authors provide a well-rounded summary of the various aspects of chronic rhinosinusitis, specifying multiple features of this inflammatory disease affecting the nasal and sinus mucosa. However, regarding biofilms—communities of microorganisms encased in a protective matrix—while the authors note that Staphylococcus aureus is the most frequently identified pathogen in CRS patients, it is important to emphasize more clearly that S. aureus biofilms contribute significantly to disease persistence and treatment failure. To this end, I suggest the authors cite the following relevant literature:

-Int J Pharm. 2023;631:122492. doi:10.1016/j.ijpharm.2022.122492

-Nat Rev Microbiol. 2022;20(10):621-635. doi:10.1038/s41579-022-00682-4

Query#2

Regarding Figure 2, which illustrates the biofilm production capacity of Staphylococci isolated from patients with CRSsNP and CRSwNP—categorized as weak biofilm producers (a), moderate biofilm producers (b), and strong biofilm producers (c)—it would be better to place panels A and B on the same line, positioned above panel C.

Comments on the Quality of English Language

Review the English form of the whole article different parts need careful checking of the English form and language.

Author Response

Comment 1. In the introduction paragraph, the authors provide a well-rounded summary of the various aspects of chronic rhinosinusitis, specifying multiple features of this inflammatory disease affecting the nasal and sinus mucosa. However, regarding biofilms—communities of microorganisms encased in a protective matrix—while the authors note that Staphylococcus aureus is the most frequently identified pathogen in CRS patients, it is important to emphasize more clearly that S. aureus biofilms contribute significantly to disease persistence and treatment failure. To this end, I suggest the authors cite the following relevant literature:

-Int J Pharm. 2023;631:122492. doi:10.1016/j.ijpharm.2022.122492

-Nat Rev Microbiol. 2022;20(10):621-635. doi:10.1038/s41579-022-00682-4

Response 1. We thank the reviewer for the insightful and useful comments of our work. We used this positive feedback to further improve the clarity and scientific tone of our paper.

In the revised manuscript, we have expanded this section to highlight the central role of S. aureus biofilms in driving persistence, antibiotic tolerance, and immune evasion, which are key factors of poor treatment outcomes in CRS. We have also incorporated the recommended references to strengthen this point and provide additional context from the current literature.

Comment 2. Regarding Figure 2, which illustrates the biofilm production capacity of Staphylococci isolated from patients with CRSsNP and CRSwNP—categorized as weak biofilm producers (a), moderate biofilm producers (b), and strong biofilm producers (c)—it would be better to place panels A and B on the same line, positioned above panel C.

Response 2. The panels of Figure 2 were positioned as required.

Round 2

Reviewer 1 Report

Comments and Suggestions for Authors

The authors have fully revised the manuscript in accordance with the comments and replied to reviewers' concerns. This revision has significantly improved the manuscript's quality, and the issues of poor presentation have been addressed.

It is recommended that the editorial team make the final decision after considering the opinions of other reviewers.